

# Testing Kubo formula on a nonlinear quantum conductor driven far from equilibrium, via power exchanges

**Zubair Iftikhar, Jonas Müller, Yuri Mukharsky, Philippe Joyez, Patrice Roche and Carles Altimiras⋆**

Université Paris-Saclay, CEA, CNRS, SPEC 91191 Gif-sur-Yvette Cedex, France

⋆ carles.altimiras@cea.fr

## Abstract

We present an experimental test of Kubo formula performed on a nonlinear quantum conductor, a Superconductor-Insulator-Superconductor tunnel junction, driven far from equilibrium by a DC voltage bias. We implement the proposal of Lesovik & Loosen [1] and demonstrate experimentally that it is possible to extract both the emission and absorption noise of the conductor by measuring the power it exchanges with a linear detection circuit whose occupation is tuned close to vacuum levels. We then compare their difference to the real part of the admittance which is independently measured by coherent reflectometry, finding that Kubo formula holds within experimental accuracy. Last, we show theoretically that the spectral density of power exchanged between a quantum conductor and its linear detection circuit follows a Lesovik & Loosen like formula, even in the presence of strong detection back-action. This result applies as long as the conductor acts as a current source for the detection circuit and the detection circuit is not singular.

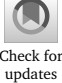

# 1  Introduction

The current flowing through a quantum conductor fluctuates even when the circuit is at rest [2, 3]. Contrary to classical stochastic signals such as thermal noise [4, 5], shot-noise [6] or $1/f$ [7] noise, the power spectral density (PSD) of quantum noise is different at positive and negative frequencies [2, 8, 9]. Remarkably, Kubo formula [10] expressed in frequency (Fourier) space relates such frequency asymmetry in the noise PSD to the dissipative linear response of the conductor when driven by a small classical external bias. Initially derived for weakly perturbed systems, Kubo formula has been experimentally tested [11] by measuring the thermal equilibrium fluctuations of linear RF resonators using a photon–assisted tunneling spectrometer [12–16]. Yet, variational predictions of the response of quantum systems to linear external drives [17, 18] show that the structure of Kubo formula holds even when the system is nonlinear and is driven arbitrarily far from equilibrium.

Here we provide an experimental test of Kubo formula by measuring both the current fluctuations and the admittance of a nonlinear quantum conductor: a superconducting-insulator-superconducting (SIS) tunnel junction, driven far from equilibrium by a DC voltage bias. We implement the proposal of Lesovik and Loosen [1, 19] and demonstrate experimentally that it is possible to extract both the positive and negative frequency components of the PSD of the SIS junction current fluctuations, by measuring the electromagnetic power it exchanges with a linear detection circuit whose occupation is tuned close to vacuum level. The difference of the two PSD obtained this way is then compared to the real part of the conductor's admittance which is independently measured by standard RF coherent reflectometry, demonstrating that Kubo formula holds within our experimental accuracy. Our results clarify that i) non-symmetrized current correlations can be extracted from real-valued measurement outputs performed on different experimental situations, but also that ii) Kubo relation holds for strongly nonlinear conductors driven far from equilibrium. We finally show that the approach

of Lesovik and Loosen can be generalized to describe the physics of quantum conductors in possibly strong interaction with their detection circuit, provided that the conductors act as good current sources to the circuit. In this scenario, Kubo formula connects the power dissipated within the conductor to its linear response, accounting for its Joule effect.

## 2 Non–symmetrized current fluctuations from power exchanges

### 2.1 Physical principle

Previous attempts at measuring both the positive and negative frequency PSD of the non–equilibrium current fluctuations flowing through nonlinear conductors observed that they are markedly different [11, 20]. However, the symmetric schemes used to couple the measured conductor to its spectrometer imprinted too much detection back-action on the measured conductor via photon–assisted transport effects, preventing the authors from extracting quantitative values. In order to avoid such back-action, we will follow a different approach, initially proposed by Lesovik and Loosen [1,19,21], based on the measurement of the power exchanged between the quantum conductor and a linearly coupled harmonic oscillator. We consider the case shown in Fig. 1 (a) where a quantum conductor with current operator $\hat{I}(t)$ is coupled to a linear detection circuit of impedance $Z_{det}$ via the QED coupling $\hat{H}_{QED} = \hat{I}\hat{\Phi}$, where the electromagnetic flux $\hat{\Phi}(t) = \int_{-\infty}^{t} dt' \hat{V}(t')$ is related to the voltage operator $\hat{V}$ at the coupling node. As shown in Appendix A, the spectral density of the power exchanged with the detection circuit $\Delta P$ around the detection frequency $f_0$ in a vanishing detection bandwidth $\Delta f$ can be computed to the lowest order in this coupling, giving:

$$\frac{\Delta P}{\Delta f} = \frac{S_{VV}(f_0)S_{II}(-f_0) - S_{VV}(-f_0)S_{II}(f_0)}{hf_0}, \tag{1}$$

where $S_{XX}(f) = \int d\tau e^{2i\pi f\tau} \langle \hat{X}(t+\tau)\hat{X}(t)\rangle$ are the PSD of the non-symmetrized time–correlation functions of the observables $\hat{X}$. The expectation values entering in (1) are taken in the interaction picture. This perturbative expression is well justified when the impedance of the quantum conductor is way larger than the impedance of the detection circuit [21]. In this limit, the quantum conductor imposes its current fluctuations on the detection impedance (current source limit), while the detection impedance imposes its voltage fluctuations on the quantum conductor at the coupling node (voltage source limit). The quantum conductor and the detection impedance behave both as sources and loads of fluctuations with respect to their surrounding circuit, explaining the symmetric role played by the spectral densities $S_{II}$ and $S_{VV}$ in Eq. (1). This expression also shows that, with our choice of Fourier transform convention, negative frequency fluctuations trigger the emission of power into the detection circuit while positive fluctuations trigger the absorption from it, justifying the names of emission and absorption noise [1, 13, 19]. For a linear detection circuit with negligible bandwidth $\Delta f$ around $f_0$, its voltage PSD can be expressed in terms of the corresponding photon occupation $n = \langle a^\dagger a \rangle$ [2,8], simplifying equation (1) as follows:

$$\frac{\Delta P}{\Delta f} = 2\operatorname{Re} Z_{det}(f_0)[(1+n)S_{II}(-f_0) - nS_{II}(f_0)], \tag{2}$$

which is the result originally obtained by Lesovik and Loosen [1]. The two positive terms describe spontaneous and stimulated emission of electromagnetic power into the detection impedance, while the last negative term describes stimulated absorption from it. The exchanged power thus contains a linear yet non-symmetric combination of the current fluctuations at its origin. Therefore, measuring the exchanged power at different calibrated occupations $n$ provides a protocol to separately extract the emission ($f < 0$), and the absorption

($f > 0$) noise of the quantum conductor. Importantly, the power exchanged with a detection circuit with vanishing occupation $n = 0$ is directly proportional to the emission noise [22].

## 2.2 Experimental implementation

Figure 1 (b) shows a simplified scheme of the circuit we use to measure such power exchanges. An SIS junction (in green) is connected to an RF resonator (implemented as a cavity filter in blue) in a dilution refrigerator of base temperature $T \simeq 15$ mK. The junction is obtained from thermal oxidation of aluminum thin films, evaporated using the double–angle evaporation technique [23], in a superconducting quantum interference device (SQUID) [24] geometry of two nominally identical junctions with a total parallel tunneling resistance $R_T = 6.7$ kΩ. The coupled resonator is a commercial third–order cavity filter, with center frequency $f_0 = 6.8$ GHz and bandwidth $\Delta f = 660$ MHz. It is 50 Ω matched in the coupled bandwidth while having vanishing impedance at other frequencies. As a result of the strong impedance mismatch $Z_{det}/R_T \simeq 1/100$, the SIS junction acts as a current source to the detection circuit. Although vanishing at the base temperature, the photon occupation $n$ in the detection bandwidth can be externally tuned with the help of an additional noise source: a Normal–Insulator–Normal tunnel (NIN) junction (in orange). The NIN junction is made similarly from aluminum thin films, but is placed in the close vicinity ($\simeq$ mm) of a neodymium magnet and shows no trace of superconducting features. The NIN junction is used as a calibrated [25–27] source of incoherent [28–30] RF shot-noise power, tuning the resonator occupation number $n$. The NIN junction of resistance $R_{NIN} = 41.8$ Ω is nearly optimally matched to the 50 Ω RF circuit which couples it non–reciprocally to the cavity filter via an RF circulator. The RF power leaking out from the cavity filter is guided through two circulators to an amplification chain adding a noise level of about 2.9 K, and is measured at room temperature with a calibrated square law detector (in red). Both junctions are DC biased via the inductive port of the bias tees used to connect them to the detection circuit. The SIS junction is shunted by a cold 50 Ω resistor, in order to obtain a stable DC voltage biasing scheme, while the NIN junction is current–biased. In the measurements shown in the article, a magnetic field of about 15 mT is applied to the SIS junction in order to i) frustrate the SQUID making its direct supercurrent negligible [24] and ii) provide some superconducting depairing [31] in order to smoothen the quasiparticle BCS peak which efficiently removes the back-bending instability of the voltage polarization at biases close to twice the superconducting gap $\Delta$ (see e.g. Appendix A in [32]). The corresponding non-linear DC conductance $dI/dV(V_{SIS})$ is shown in Figure 1 (c), it is obtained from the measurement of the differential resistance of its parallel composition with the 50 Ω biasing resistor as explained in Appendix E. It mainly displays features of quasiparticle tunneling showing a DC current transport gap at $eV_{SIS} = 2\Delta = 400$ μeV, with negligible contribution from the Josephson effect.

## 2.3 Experimental results

Figure 2 shows the power exchanged between the SIS junction and the detection circuit via the cavity filter of frequency $f_0 = 6.8$ GHz, as a function of the SIS voltage bias $V_{SIS}$ for a given voltage bias $V_{NIN}$ applied to the NIN junction. It is defined as the difference between the power measured by the square law detector (red diode symbol in Figure 1 (b)) at finite SIS bias $V_{SIS}$ and the power measured at zero SIS bias:

$$\Delta P_{exch} = P(V_{SIS}, V_{NIN}) - P(V_{SIS} = 0, V_{NIN}). \tag{3}$$

For each curve, the exchanged power is shown normalized with respect to the amplification chain noise $P(V_{SIS} = 0, V_{NIN} = 0) = k_B T_{amp} \Delta f$ correcting for slow gain drifts of the room–temperature detection chain. Let us stress that each of the power measurements on the right-

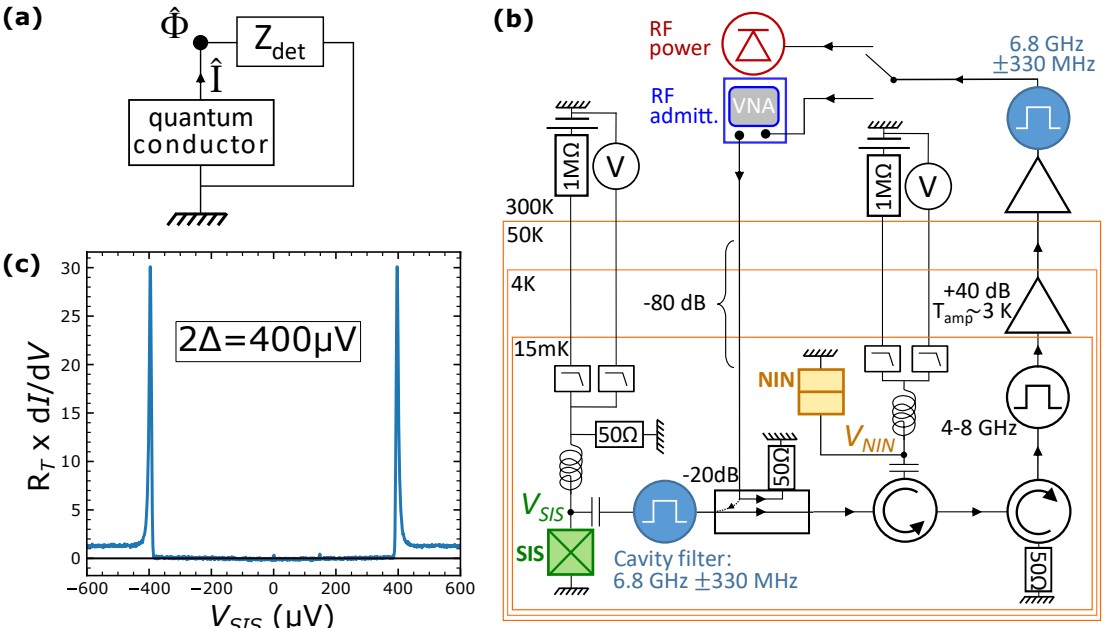

Figure 1: **(a)** A quantum conductor with current operator $\hat{I}$ is galvanically coupled to a linear circuit with impedance $Z_{det}$. The minimal QED coupling reads $\hat{H}_{QED} = \hat{I}\hat{\Phi}$, where $\hat{\Phi}$ is the electromagnetic flux $\hat{\Phi}(t) = \int_{-\infty}^{t} dt' \hat{V}(t')$ defined at the coupling node and $\hat{V}$ the corresponding voltage. **(b)** Scheme of the circuit used to test Kubo formula. The quantum conductor to be measured is a SIS junction (in green) connected to the circuit via a bias tee. Its inductive port is used to DC-voltage bias and measure it. The capacitive port is used to couple it to a RF detection circuit via a 660 MHz bandwidth cavity filter centered at $f_0 = 6.8$ GHz (in blue) whose impedance is vanishing out of the coupled band. The photon occupation of this narrow band detection impedance can be externally tuned by the RF shot–noise emitted by a 50 Ω matched DC–biased NIN junction (in orange) coupled non-reciprocally to the cavity via a 18 dB isolation circulator. The noise power contained in the cavity filter is routed with a pair of circulators to a cryogenic amplifier and detected at room temperature with a square law detector (red diode symbol). A −20 dB directional coupler inserted after the cavity filter is used to shine a heavily attenuated RF tone generated by a VNA at the input of the SIS junction. The reflected signal is routed to the VNA detection port after amplification in order to measure the SIS admittance. **(c)** $dI/dV(V_{SIS})$ curve of the SIS junction obtained from the dV/dI measured in a three point configuration at its 50 Ω shunted input. The tunneling resistance obtained is $R_T = 6.7$ kΩ and the superconducting gap is $2\Delta = 400$ μeV.

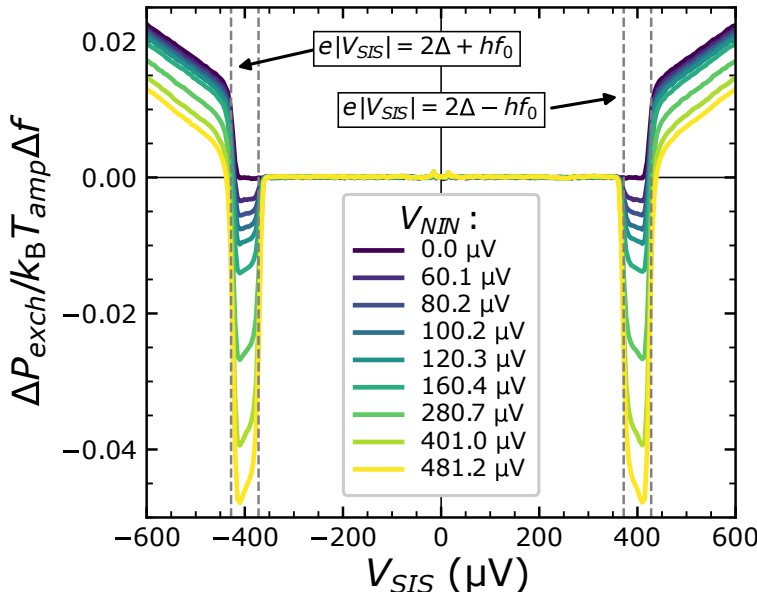

Figure 2: Power exchanged between the SIS junction and its linear detection circuit within the coupled bandwidth $\Delta f = 660\,\text{MHz}$ around frequency $f_0 = 6.8\,\text{GHz}$. It is defined by Eq. 3 and is normalized by the amplification chain noise $k_{\text{B}} T_{amp} \Delta f = P(V_{SIS} = V_{NIN} = 0)$ (roughly 2.9 K noise temperature). At zero NIN bias, the exchanged power is always positive: the SIS junction can only emit into the detection circuit. The onset for this spontaneous emission is found at $eV_{SIS} = 2\Delta + hf_0$, where $hf_0 = 28\,\mu\text{eV}$ is the energy quantum of the cavity filter. At finite NIN voltage bias $V_{NIN}$, the linear circuit is driven out of equilibrium by the incoherent radiation emitted by the NIN junction providing it with finite photon occupation. In this case the exchanged power can also be negative, meaning the SIS junction is absorbing some power from the detection circuit. The onset for this power absorption is $eV_{SIS} = 2\Delta - hf_0$, while the exchanged power becomes positive again after the emission threshold.

hand side of Eq. (3) corresponds to the RF noise being rectified by the square law detector, and according to quantum network theory [33] it is proportional to the *symmetrized* PSD of the voltage fluctuations at the input of the detector, see e.g. [34] or the Appendix B. However, their difference describes how much the photon occupation changes when the RF field interacts with the SIS junction at finite $V_{\text{SIS}}$ bias with respect to the zero bias $V_{\text{SIS}} = 0$, case where the SIS junction acts as an open circuit leaving the photon occupation untouched. The corresponding changes in the power measured in the detection circuit are precisely what is predicted by Eqs. (1) and (2), as we will now demonstrate experimentally.

At zero NIN bias, the detection circuit has a vanishing equilibrium photon occupation in the coupled bandwidth. In this case, we observe a strictly positive power exchange meaning that the SIS junction is **emitting** energy into the cavity. We also observe a marked SIS bias onset for such power emission at $|eV_{SIS}| = 2\Delta + hf_0$, which is in agreement with microscopic predictions [35,36] and measurements [11,20] of the SIS junction **emission noise**. Note that inelastic tunneling effects [22,37–39] are made negligible by construction, thanks to the small impedance of the detection circuit $Z_{det} = 50\,\Omega \ll h/e^2$. At finite voltage biases $V_{\text{NIN}}$ applied to the NIN junction, the incoherent radiation it emits into the circuit increases the resonator's occupation $n(V_{\text{NIN}})$. In this case, we observe a negative dip of exchanged power in the SIS

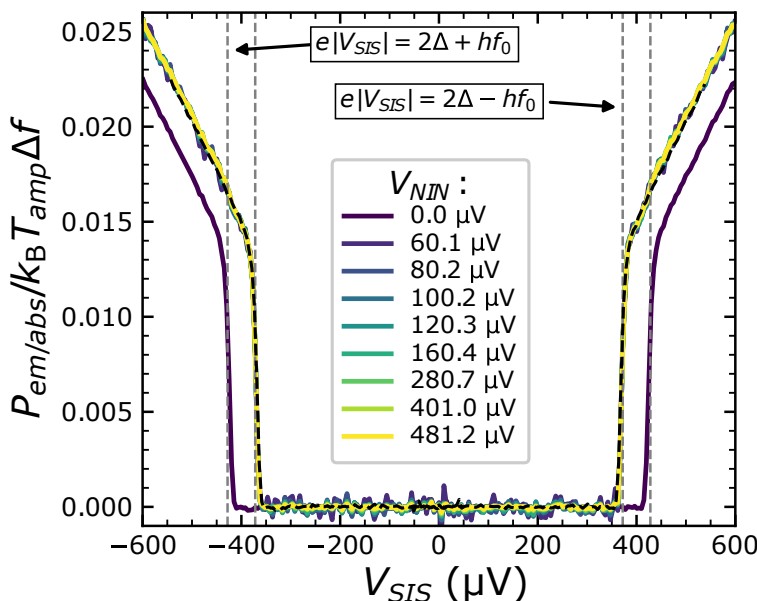

Figure 3: Emission and absorption noise power at frequency $f_0 = 6.8$ GHz of the SIS junction extracted from the power exchanges shown in Figure 2 exploiting Eq. (2) and the protocol described in the article: The emission power is directly obtained from the zero occupation ($V_{NIN} = 0$) measurement according to Eqs. (4), while the absorption noise is obtained using Eq. (5) for all the power exchanges measured at finite occupation $n(V_{NIN})$ ($V_{NIN} \neq 0$). All absorption noise curves measured at different NIN biases collapse on the same shape. The black dashed curve is obtained by applying $\pm 2hf_0 = 56$ µeV horizontal offsets to the emission power curve at negative/positive biases, which according to Rogovin and Scalapino [35] should reproduce the absorption noise in the $k_B T \ll eV_{SIS}$ limit.

voltage range $2\Delta - hf_0 < |eV_{SIS}| < 2\Delta + hf_0$ meaning that the SIS junction is **absorbing** energy from the resonator. The lower bound $|eV_{SIS}| = 2\Delta - hf_0$ corresponds to the onset of **absorption** noise in SIS junctions, in agreement with microscopic theory [35, 36]. Since the SIS junction cannot emit RF power when $|eV_{SIS}| < 2\Delta + hf_0$, the observed negative dip should be proportional to its absorption noise only and to the occupation number. Indeed, the amplitude of this dip is seen to increase with the applied NIN voltage which increases the occupation number in the coupled bandwidth.

These observations are in qualitative agreement with Eq. (2), we now use it to quantitatively extract both the emission $P_{em}$ and absorption noise $P_{abs}$ power of the SIS junction. The protocol is the following; first we identify the emission noise as the noise power exchanged at zero NIN voltage and then we invert Eq. (2) to isolate the absorption noise contribution at finite NIN voltage:

$$P_{em} = \Delta P_{exch}(V_{SIS}, V_{NIN} = 0), \tag{4}$$

$$P_{abs} = \frac{\left(1 + n(V_{NIN})\right)P_{em} - \Delta P_{exch}(V_{SIS}, V_{NIN})}{n(V_{NIN})}. \tag{5}$$

This protocol requires calibrating the occupation number within the cavity filter's bandwidth. In order to do so, we exploit the noise–power emitted by both SIS and NIN junctions in the classical limit $|eV| \gg hf_0$, where the emitted current shot-noise has a white PSD whose integrated power increases with the applied DC bias $I_{DC}$ according to a Schottky formula $\delta I^2 = 2eI_{DC}\Delta f$

see e.g. [25–27,40]. Thanks to such known noise sources, we can calibrate the attenuation of the different RF paths of our detection chain and infer the occupation of the cavity filter as a function of the NIN bias $n(V_{NIN})$ as detailed in the Appendix D. Figure 3 shows the emission and the absorption noise power of the SIS junction extracted by this procedure as a function of the SIS bias voltage. It is remarkable that all absorption noise curves constructed from data measured at different, yet finite, photon occupations collapse into the same curve and depend only on $V_{SIS}$, validating the protocol. Most importantly, it shows that Lesovik and Loosen's [1] formula Eq. (2) accurately describes the power exchanges between the junction and its detection circuit in the current source limit $Z_{det}/R_T \ll 1$. The obtained emission and absorption noise powers have a shape similar to the $I(V)$ curves of SIS junctions [36,41], but with a voltage shift of $-hf_0/e$ and $hf_0/e$, respectively. This is a general property of weakly coupled (tunneling) conductors for which it is possible to derive non-equilibrium fluctuation dissipation relations [35,38,42,43] relating the current fluctuations or the finite frequency admittance to the DC $I(V)$ curve.

We stress, that despite our calibration efforts, the data shown in Figure 3 is obtained with an additional correction to the calibrated occupation of 20% (0.8 dB) to obtain a better agreement with microscopic predictions as shown in more detail in Appendix D. This discrepancy most probably arises from the small yet finite spurious wave reflections at the input of the several RF components placed between the SIS and the NIN junctions, which are not accounted for in our calibration procedure.

## 3  Experimental test of Kubo formula

Having measured the emission and absorption current PSD we are now in a position to test Kubo formula [10]. The frequency (Fourier) space representation of Kubo formula relates the linear response to an external coherent drive $V(t) = \delta V_{ac} \cos(2\pi f t)$ as defined in Eq. (6) to the spontaneous fluctuations already present when $\delta V_{ac} = 0$. More specifically, the difference between the absorption and emission noise spectral densities of current fluctuations is predicted to be proportional to the in-phase response of the quantum conductor Eq. (7), while the out-of-phase response is obtained from it according to Kramers-Kronig relations Eq. (8):

$$\frac{\delta I(t)}{\delta V_{ac}} = \operatorname{Re} Y(f)\cos(2\pi f t) + \operatorname{Im} Y(f)\sin(2\pi f t), \tag{6}$$

$$\operatorname{Re} Y(f) = \frac{S_{II}(f) - S_{II}(-f)}{2hf}, \tag{7}$$

$$\operatorname{Im} Y(f) = -\frac{2}{\pi}\mathcal{P}\int_0^\infty df' \frac{f \operatorname{Re} Y(f')}{f'^2 - f^2}. \tag{8}$$

Initially derived for systems sitting close to equilibrium, Svirskii and co-workers [17] stressed that the same structure remains valid for arbitrary stationary out-of-equilibrium situations, albeit the correlation functions must then be computed in the non-equilibrium steady state (see [18] for a generalization to nonstationary drives).

In order to test Kubo formula Eq. (7), we need to measure the linear response of the conductor to a single–tone excitation while driven far from equilibrium by the DC bias. As shown in Figure 1 (b), we inserted for this purpose a $-20$ dB directional coupler in the detection chain. It is used to send a single-tone excitation via the coupled port while the reflected signal is routed via the detection chain back to the Vector Network Analyzer (VNA), which measures the in-phase and out-of-phase amplitudes of the signal reflected by the junction according to its linear response Eq. (6). The $-5$ dBm power level delivered by the VNA is lowered by 80 dB

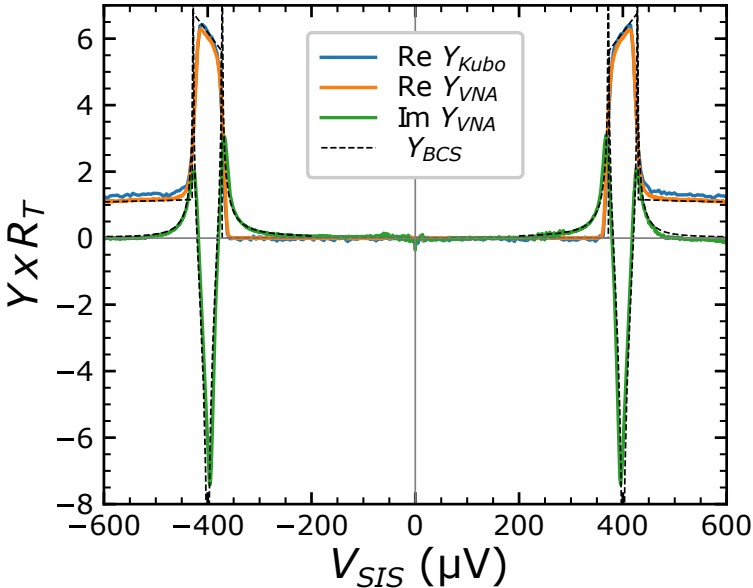

Figure 4: Admittance of the SIS junction as a function of DC bias measured in the coupled band of the cavity filter. Blue line: real part of the admittance of the SIS tunnel junction Re $Y_{Kubo}$ extracted from the emission and absorption noise spectral densities shown in Figure 3 using Kubo formula Eq. (7). It is compared to the real (orange line) and imaginary (green line) parts of the admittance independently measured by coherent reflectometry with a Vector Network Analyzer $Y_{VNA}$ using Eq. (9). Dashed lines are the corresponding predictions from BCS theory in the zero temperature and zero depairing limit detailed in the Appendix C.2.

with matched attenuators thermally anchored to the different stages of the dilution fridge. After the directional coupler, this results in an excitation level of about $-105\,\text{dBm} = 1.2\,\mu\text{V}$ at the input of the SIS junction. This level is low enough to prevent nonlinear photon-assisted transport effects [12, 44] $e\delta V_{rms}/hf \simeq 4\%$. Of course, one needs to calibrate the gain of the measurement chain (in both amplitude and phase), in order to extract the response of the junction quantitatively. Our calibration procedure, detailed in Appendix C, assumes that i) for the largest applied bias ($V_{SIS} = -600\,\mu\text{V}$) the SIS junction acts, as predicted from BCS theory, essentially as a plain resistor with an effective resistance slightly different from the tunneling resistance $6.7\,\text{k}\Omega/1.1$, ii) the junction behaves as an open circuit when DC biased in about the middle of the transport gap $V_{SIS} = -225\,\mu\text{V}$, and iii) the detection impedance seen by the junction does not depend on the junction's admittance. With these assumptions, we can extract the SIS junction admittance from its measured reflection coefficient $\Gamma(f)$:

$$Y_{VNA}(f)Z_{det}(f) = \frac{1-\Gamma(f)}{1+\Gamma(f)}. \tag{9}$$

Figure 4 shows the real (in orange) and imaginary (in green) parts of the admittance $Y_{VNA}$ extracted from this method, averaged over 13 frequency values equally distributed between $f = 6.5\,\text{GHz}$ and $f = 7.1\,\text{GHz}$ with 50 MHz steps, fully covering the coupled bandwidth of the cavity filter. This averaged admittance is compared to the real part of the admittance $Y_{Kubo}$ as dictated by Kubo formula (7) using the emission and absorption noise measurements

performed on the $6.5 - 7.1$ GHz coupled bandwidth shown in Figure 3, namely

$$4hf \operatorname{Re} Z_{det} \operatorname{Re} Y_{Kubo} = \frac{P_{abs} - P_{em}}{k_B T_{amp} \Delta f} \frac{k_B T_{amp}}{\alpha} \,.$$

The first term on the right–hand side is simply the difference between the absorption and the emission noise shown in Figure 3, whereas the second term is the noise power of the detection chain $k_B T_{amp}$ referred to the output of the SIS junction via a power attenuation coefficient $\alpha = 0.717$ calibrated in the Appendix D. Note that according to our calibration procedure we have by construction $\operatorname{Re} Y_{VNA}(-600\,\mu V) = \operatorname{Re} Y_{BCS}(-600\,\mu V)$ but also that $\operatorname{Im} Y_{VNA}(-600\,\mu V) = 0$. Nevertheless, the curves agree throughout the probed voltage range where the admittance varies by a factor of $\simeq 6$ validating the experimental protocols. Most importantly, the agreement obtained between $\operatorname{Re} Y_{Kubo}$ and $\operatorname{Re} Y_{VNA}$ confirms experimentally the validity of Kubo formula Eq. (7) for a strongly nonlinear conductor driven far from equilibrium.

## 4 Joule heating and Kubo formula

The original derivation of Lesovik and Loosen formula Eq. (2) assumes a weak coupling between the conductor and the resonator. Therefore, the current-correlation functions appearing in Eq. (2) are meant to be taken in the noninteracting limit as detailed in the Appendix A. However experiments [22, 38, 45] measuring the power emitted by tunnel junctions interacting strongly with a linear detection circuit, giving rise to strong back-action effects (*aka* Dynamical Coulomb blockade [37]), showed that Eq. (2) could still be used in the vanishing occupation limit in agreement with perturbative calculations with respect to the tunneling Hamiltonian [21, 45]. In this case, the current-correlation functions describing the emitted power were computed including the strong back-action effects arising from the environment.

Here we show that, owing to the linearity of the electromagnetic detection circuit, we can generalize Eq. (2) in order to take into account strong back-action effects. This new derivation computes the spectral density of power exchanged within a vanishingly small bandwidth, it is valid provided i) the detection circuit is not singular and ii) there is a strong impedance mismatch such that the conductor is in the current source limit. The key point is that both the electromagnetic environment Hamiltonian and the electromagnetic flux $\hat{\Phi}$ appearing in the QED coupling can be linearly expanded in terms of the electromagnetic modes described by the bosonic fields $\hat{a}(f)$:

$$\hat{H}_{EM} = \int_0^{+\infty} df\, hf(\hat{a}^\dagger(f)\hat{a}(f) + 1/2) = \int_0^{+\infty} df\, \hat{h}(f)\,,$$

$$\hat{\Phi} = \int_0^{+\infty} df\, \sqrt{\frac{h \operatorname{Re} Z(f)}{2\pi^2 f}}(\hat{a}(f) + \hat{a}^\dagger(f)) = \int_0^{+\infty} df\, \hat{\phi}(f)\,.$$

One can single out a detection mode of frequency $f_0$ with vanishing detection bandwidth $\Delta f / f_0 \ll 1$ from the full electromagnetic environment:

$$\hat{H}_{EM} = \hat{H}_{EM}^{\backslash} + \hat{h}(f_0)\Delta f\,,$$

$$\hat{\Phi} = \hat{\Phi}^{\backslash} + \hat{\phi}(f_0)\Delta f\,.$$

If the electromagnetic environment has a continuous detection impedance $\operatorname{Re} Z_{det}(f)$ which is characteristic of open quantum systems, and its state has non-singular dynamics in the vicinity of the detection mode $\langle \hat{h}(f_0 + \Delta f) \rangle \simeq \langle \hat{h}(f_0) \rangle$, then whatever correlation function of the

truncated field $\hat{\Phi}^{\backslash}$ computed within the dynamics dictated by the truncated Hamiltonian $\hat{H}^{\backslash}_{EM}$ will tend to those of the full Hamiltonian in the vanishing detection bandwidth limit, e.g.:

$$\lim_{\Delta f/f_0 \to 0} \langle \hat{\Phi}^{\backslash}(t)\hat{\Phi}^{\backslash}(0) \rangle_{\hat{H}^{\backslash}_{EM}} = \langle \hat{\Phi}(t)\hat{\Phi}(0) \rangle_{\hat{H}_{EM}}. \tag{10}$$

By singularizing the detection mode of frequency $f_0$ from the full Hamiltonian

$$\begin{aligned}
\hat{H} &= \hat{H}_{conductor} + \hat{H}^{\backslash}_{EM} + \hat{I}\hat{\Phi}^{\backslash} + \Delta f(\hat{h}(f_0) + \hat{I}\hat{\phi}(f_0)) \\
&= H_I + \Delta f(\hat{h}(f_0) + \hat{I}\hat{\phi}(f_0)),
\end{aligned} \tag{11}$$

it is apparent that in the vanishing detection bandwidth limit its coupling to the quantum conductor can be treated as a perturbation to an interaction picture Hamiltonian $H_I$ which takes into account the (possibly strong) interaction between the quantum conductor and the rest of the electromagnetic environment. Note, however, that such an expansion is strictly valid only if larger order terms are parametrically smaller and thus that $\mathrm{Re}\, Y_{conductor}(f_0)\,\mathrm{Re}\, Z(f_0) \ll 1$ [21]. It follows then that in this limit the spectral density of power exchanged between the quantum conductor and its detection mode, defined as $S_P(f) = \lim_{\Delta f \to 0} \frac{\Delta P}{\Delta f}$, follows a Lesovik & Loosen formula:

$$\begin{aligned}
\frac{S_P(f, n(f))}{2\,\mathrm{Re}\, Z_{\mathrm{det}}(f)} &= (1 + n(f))S^{\backslash}_{II}(-f) - n(f)S^{\backslash}_{II}(f) \\
&= S_{II}(-f) - 2\,\mathrm{Re}\, Y(f)hf\,n(f).
\end{aligned}$$

In the second equality, we exploited the fact that the current correlations computed in the interaction picture provided by $\hat{H}_I$: $S^{\backslash}_{II}(f)$ tend to those computed using the full Hamiltonian Eq. (11) in the vanishing detection bandwidth limit. We also exploited Kubo formula [10, 17, 18] to factorize the terms proportional to the occupation number. It follows that the power exchanged due to the presence of a finite occupation $n$ reads:

$$S_P(f, n) - S_P(f, 0) = -4\,\mathrm{Re}\, Z_{\mathrm{det}}(f)\,\mathrm{Re}\, Y(f)hf\,n,$$

which demonstrates that, whatever complicated dynamics results from the full QED coupling, the conductor dissipates the energy $hf\,n(f)$ contained within a narrow band $\Delta f$ around frequency $f$ of its detection circuit at a rate given by the real part of the conductor's admittance. In other words combining both Kubo and Lesovik & Loosen's formulas accounts for the Joule power dissipated within the conductor, in the limit where the external circuit behaves as a voltage source to the conductor. Note also that if the conductor is driven in an out-of-equilibrium state such that its admittance becomes negative, then the net exchanged power $S_P(f, n) - S_P(f, 0)$ becomes positive and the conductor acts as an amplifier. In this case, Kubo relation describes the gain of the system in the linear response regime [46].

## 5 Conclusion

Summing up, we presented an experimental test of Kubo formula Eq. (7) for a nonlinear conductor driven far from equilibrium by a stationary voltage bias. We first demonstrated that both emission and absorption current noises of the conductor can be extracted from the power it exchanges with a narrow–band linear detection circuit with a calibrated occupation (close to vacuum levels), in agreement with Lesovik and Loosen's formula Eq. (2). The circuit was designed in the strong impedance mismatch limit ($\mathrm{Re}\, Y Z_{det} \ll 1$) such that the energy exchanges can be described by such a simple formula. This result demonstrates that non–symmetrized

correlation functions can be extracted from the comparison of power measurements performed in different experimental conditions. We then measured the conductor's admittance through phase–sensitive reflectometry measurements performed in the linear response regime. The real part of the admittance obtained this way could be compared to the difference between the absorption and the emission noise measured from power exchanges. We found that Kubo formula Eq. (7) agrees with our data within our experimental accuracy, even though the conductor has a strongly nonlinear $I(V)$ characteristic and is driven far from equilibrium. Last, we argued that Lesovik and Loosen formula Eq. (2) describes quite generically the spectral density of power exchanged between the conductor and its surrounding circuit even in the presence of strong detection back-action effects, provided the circuit is not singular and the conductor is in the good current source limit with respect to its detection circuit. Such conditions on the environment dynamics apply thus quite generally to open linear detection circuits. However, if the conductor and its detection circuit are not strongly impedance mismatched, a sizable voltage drop develops at the input of the conductor proportional to the current it injects into the detection circuit. As a consequence, the (nonlinear) conductor becomes part of its own environment and Lesovik and Loosen formula looses its validity. Even though exact results [47–49] exist describing particular circuits [50, 51] or self-consistent Gaussian approximation schemes can be devised [52], we are not aware of a simple general way of handling the corresponding nonlinear out-of-equilibrium physics to be solved self-consistently.

## Acknowledgments

We gratefully acknowledge stimulating discussions within the Quantronics and Nanoelectronics groups.

**Funding information**   This work received funding from the European Research Council (Horizon 2020/ERC Grant Agreement No. 639039 'NSECPROBE'), from the French ANR (contract 'SIMCIRCUIT' ANR-18-CE47-0014-01) and from the German-French ANR/DFG grant 'StrongQEDmpc' (ANR-22-CE92-0053).

**Data availability**   The full set of data shown in this article is available in the public repository DOI: 10.5281/zenodo.15038601.

## A   Perturbative computation of power exchanges

We consider a quantum conductor with current operator $\hat{I}$ having stationary dynamics dictated by a Hamiltonian $\hat{H}_{cond}$. It is coupled to a linear detection circuit described by $\hat{H}_{EM}$. The Hamiltonian of the coupled system is:

$$\hat{H} = \hat{H}_{cond.} + \hat{H}_{EM} + \hat{I}\hat{\Phi},$$

with $\hat{\Phi}$ the electromagnetic flux at the coupling node such that $\hat{\Phi} = \int_{-\infty}^{t} \hat{V}(t')dt'$ and $\hat{V}$ the corresponding voltage. We wish to compute how much this coupling changes the energy initially present in the linear circuit. The power conveyed into it is by definition:

$$\hat{P}_{EM} = \frac{d\hat{H}_{EM}}{dt} = \frac{i}{h}[\hat{H}, \hat{H}_{EM}] = -\hat{I}\hat{V},$$

where we exploited $[\hat{H}_{cond.}, \hat{H}_{EM}] = [\hat{I}, \hat{H}_{EM}] = 0$ and $\frac{i}{\hbar}[\hat{H}_{EM}, \hat{\Phi}] = \hat{V}$. Expanding it to lowest order in the coupling $\hat{I}\hat{\Phi}$ we obtain:

$$\hat{P}_{EM} \simeq -\hat{I}_I \hat{V}_I - \frac{i}{\hbar} \int_{-\infty}^{t} dt'[\hat{I}_I(t')\hat{\Phi}_I(t'), \hat{I}_I(t)\hat{V}_I(t)], \tag{A.1}$$

where $\hat{X}_I(t)$ denotes the time dependence of observable $\hat{X}$ in the uncoupled interaction picture.

## A.1 Stationary EM states

Taking the expectation value of Eq. (A.1), the first term vanishes since $\langle \hat{V}_I \rangle = 0$ for the thermal stationary states considered in the present work. The remaining term can be recast with the help of the current $S_{II}$ and voltage $S_{VV}$ spectral densities $\langle \hat{I}(t+\tau)\hat{I}(t)\rangle = \int df\, S_{II}(f)e^{-2i\pi f\tau}$ and $\langle \hat{\Phi}(t+\tau)\hat{V}(t)\rangle = \int df\, \frac{S_{VV}(f)}{-2i\pi f}e^{-2i\pi f\tau}$ since they simply factorize in the interaction picture:

$$\begin{aligned}
\langle \hat{P}_{EM} \rangle &\simeq \frac{2}{\hbar} \mathrm{Im} \int_{-\infty}^{0} d\tau \langle \hat{I}(t+\tau)\hat{I}(t)\rangle \langle \hat{\Phi}(t+\tau)\hat{V}(t)\rangle \\
&\simeq \int_{-\infty}^{+\infty} df\, \frac{S_{II}(-f)S_{VV}(f)}{hf} \\
&\simeq \int_{0}^{+\infty} df\, \frac{S_{II}(-f)S_{VV}(f) - S_{II}(f)S_{VV}(-f)}{hf}.
\end{aligned}$$

This last equation becomes Eq. (1) in the vanishing detection bandwidth limit. Since for stationary states one can simply express voltage spectral densities in terms of photon occupation numbers $n(f) = \langle \hat{a}^{\dagger}(f)\hat{a}(f)\rangle$:

$$\begin{aligned}
S_{VV}(f) &= 2hf\, \mathrm{Re}\, Z(f)(1+n(f)), \\
S_{VV}(-f) &= 2hf\, \mathrm{Re}\, Z(f)n(f),
\end{aligned}$$

one obtains Eq. (2). The perturbative expansion Eq. (A.1) giving rise to such simple formulas neglects higher-order terms in the QED coupling $\hat{I}\hat{\Phi}$. As these terms scale as powers of the product of the detection impedance and the conductor's admittance [21], Eq. (2) is only valid in the limit where the conductor acts as a good current source to its detection circuit.

## A.2 AC bias

For the sake of completeness we discuss as well the case of a small AC bias being applied by the detection circuit. We take the detection circuit to be set in a single-tone displaced state with frequency $f_0$ such that $\langle \hat{V}_I(t)\rangle = V_{ac}\cos(2\pi f_0 t)$. Again, the expectation value of the first term in Eq. (A.1) vanishes, such as its time average:

$$\overline{\langle \hat{I}_I \hat{V}_I \rangle} = \lim_{T\to\infty} \frac{1}{T} \int_{-T/2}^{T/2} dt \langle \hat{I}_I(t)\rangle \langle \hat{V}_I(t)\rangle = 0.$$

The voltage spectral density now displays an explicit time dependence. Averaging it over the driving period generates a singular contribution at the driving frequency:

$$\begin{aligned}
\overline{S_{VV}(f,t)} &= 2hf\, \mathrm{Re}\, Z(f) + \frac{V_{ac}^2}{4}\delta(f-f_0), \\
\overline{S_{VV}(-f,t)} &= \frac{V_{ac}^2}{4}\delta(f-f_0).
\end{aligned}$$

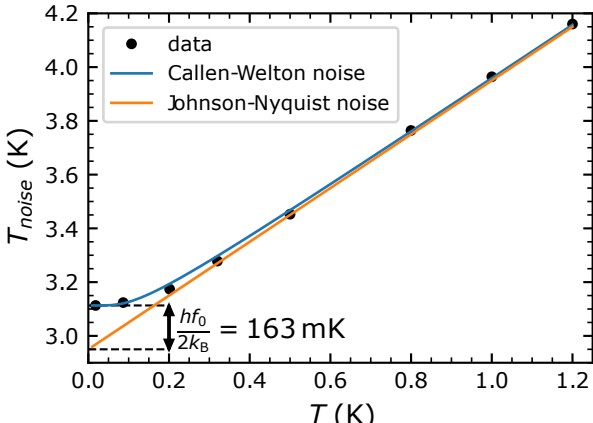

Figure 5: Noise temperature measured in the coupled bandwidth with the square–law detector as a function of the temperature set to the mixing chamber plate. The noise temperature units are extracted from fitting its linear evolution to Johnson-Nyquist formula and thus assume a perfect matching between the line and the amplifier. The data is then compared to the Callen-Welton noise prediction.

Taking the time average of the expectation value of Eq. (A.1) and exploiting Kubo formula we obtain the power being exchanged on average between the conductor and its linear circuit:

$$\overline{\hat{P}_{EM}} = -\operatorname{Re} Y(f_0)\frac{V_{ac}^2}{2} + \int_0^{+\infty} df\, 2\operatorname{Re} Z(f)S_{II}(-f).$$

This expression shows that the Joule effect within the conductor (the power it dissipates from the driving tone) is proportional to the real part of its admittance, namely:

$$\overline{\hat{P}_{EM}}(V_{ac}) - \overline{\hat{P}_{EM}}(V_{ac}=0) = -\operatorname{Re} Y(f_0)\frac{V_{ac}^2}{2}.$$

## B Thermal noise characterization

In this section, we show the thermal characterization of the amplifier used to perform the RF power measurements, demonstrating that the rectified voltages measured by the square-law detector are proportional to the symmetrized voltage noise propagating in the lines. In Figure 5 we plot the RF noise power rectified by our square-law detector in the coupled bandwidth of the cavity filter as a function of the mixing chamber plate temperature. Note that all dissipative conductors (50 Ω match of the circulator, NIN junction) connected to the detection line are thermally anchored to the mixing chamber plate. For these measurements, all voltages applied to the tunnel junctions are set to zero, and the mixing chamber plate temperature is set within the percent level by a PID controller acting on a heating resistor while monitoring the temperature. Data shown in figure 5 is the power recorded after waiting for about 50 minutes at a given stable temperature below 800 mK and about 20 minutes above 1 K. The conversion between the measured rectified voltage and the noise temperature $k_B T_{noise}$, giving the units of the vertical axis, is extracted from the linear fit to the classical Johnson-Nyquist noise power per unit bandwidth predicted for a matched detection impedance: $k_B T_{noise} = k_B T + k_B T_{amp}$ (shown as an orange line).

According to quantum network theory [33], the voltage rectified by a square-law detector should be proportional to the symmetrized spectral density of voltage fluctuations at its input

on top of the noise added by the setup. In the case of thermal radiation, the symmetrized voltage fluctuations across a circuit of impedance $Z(f)$ in an open-circuit configuration are given by Callen-Welton noise [8]:

$$
\begin{aligned}
S_{VV}(f) + S_{VV}(-f) &= 2hf \operatorname{Re} Z(f)\left(\frac{2}{e^{hf/k_B T} - 1} + 1\right) \\
&= 2hf \operatorname{Re} Z(f) \coth(hf/2k_B T).
\end{aligned}
$$

The linear cryogenic amplifier used to detect the radiation is matched to the $\operatorname{Re} Z(f) \simeq 50\,\Omega$ circuit guiding it, causing a $1/2$ voltage division at the amplifier input with respect to the above open-circuit prediction. The noise temperature referred to this matched input thus reads:

$$
\begin{aligned}
k_{\mathrm{B}} T_{noise} &= \frac{1}{4}\frac{S_{VV}(f) + S_{VV}(-f)}{\operatorname{Re} Z(f)} + k_{\mathrm{B}} T_{amp} \\
&= \frac{hf}{2}\coth(hf/2k_B T) + k_{\mathrm{B}} T_{amp},
\end{aligned}
\tag{B.1}
$$

where $k_{\mathrm{B}} T_{amp}$ is the noise added by the cryogenic amplifier [53], since it dominates the noise added by the setup owing to the large gain $\simeq 40\mathrm{dB}$ of the first amplification stage. The good agreement between Callen-Welton noise Eq. (B.1) and the data shown in Figure 5 demonstrates that the voltage rectified by the square–law detector is indeed proportional to the symmetrized voltage fluctuations at its input. Notably, the difference between Johnson-Nyquist noise and the data in the $k_{\mathrm{B}} T \ll hf$ limit is given by the noise power of zero-point motion $\frac{hf_0}{2k_{\mathrm{B}}} = 163$ mK.

However, the added noise obtained by this procedure ($T_{amp} = 2.95$ K) is 1.6 dB larger than expected from the amplifier specification sheet. This might be explained by an effective line impedance $Z(f)$ seen at the input of the amplifier larger than its nominal $50\,\Omega$ input, or simply by a non-optimal bias of the amplifier.

## C  Admittance measurement details

### C.1  Experimental protocol

Here we provide further details on the admittance measurements obtained from phase–sensitive reflectometry performed with a Vector Network Analyzer. Since the SIS junction and the detection line are strongly impedance mismatched, the signal reflected by the junction overwhelms the -20dB coherent leak of the coupler used to drive the circuit which can be neglected. We thus assume that the signal measured by the VNA can be cast in terms of the (complex) gain of the chain $G(f)$ and the reflection coefficient $\Gamma(f, V_{SIS})$ at the input of the SIS junction, namely:

$$
S_{21}(f, V_{SIS}) = \frac{V_{out}(f)}{V_{in}(f)} = G(f)\Gamma(f, V_{SIS}).
$$

Our calibration procedure assumes that, at the probed frequencies $f \in [6.5, 7.1]\,\mathrm{GHz}$, the SIS junction acts as an open circuit in the middle of its transport gap $\Gamma(f, V_{SIS} = -225\,\mu\mathrm{V}) = 1$. Therefore, the trace recorded by the VNA at this value characterizes the gain of the chain from which we can deduce the reflection coefficient:

$$
\Gamma(f, V_{SIS}) = \frac{S_{21}(f, V_{SIS})}{S_{21}(f, V_{SIS} = 225\,\mu\mathrm{V})}.
$$

From the reflection coefficient obtained this way, we extract the product $Y_{SIS}(f)Z_{det}(f)$ using Eq. (9).

If we first assume that the detection impedance is simply $Z_{det} = 50\,\Omega$, as nominally intended by making use of $50\,\Omega$ matched cables and RF-elements, we obtain the admittances shown in Figure 6 (a). The strong variations observed among these curves show that there are standing waves not captured by this simple approach.

To progress further we note that, the junction being strongly mismatched to the detection line, its reflection coefficient is essentially unity. We can thus assume that the standing wave patterns of the detection circuit do not depend on the precise value of the SIS junction admittance. However, we need to calibrate the corresponding frequency–dependent detection impedance $Z_{det}(f)$. In order to do so, we assume that the junction admittance at the largest bias measured ($eV_{SIS} = 3\Delta$) bias is real with a value given by the BCS predictions $\mathrm{Re}\,Y(eV_{SIS} = 3\Delta, hf/\Delta = 0.14) \simeq 1.1/R_T$ we detail in the next subsection. The real (blue) and imaginary (orange) parts of the calibrated detection impedance are shown in Figure 6 (d). The real part shows an oscillating pattern compatible with a 25 cm electric length standing

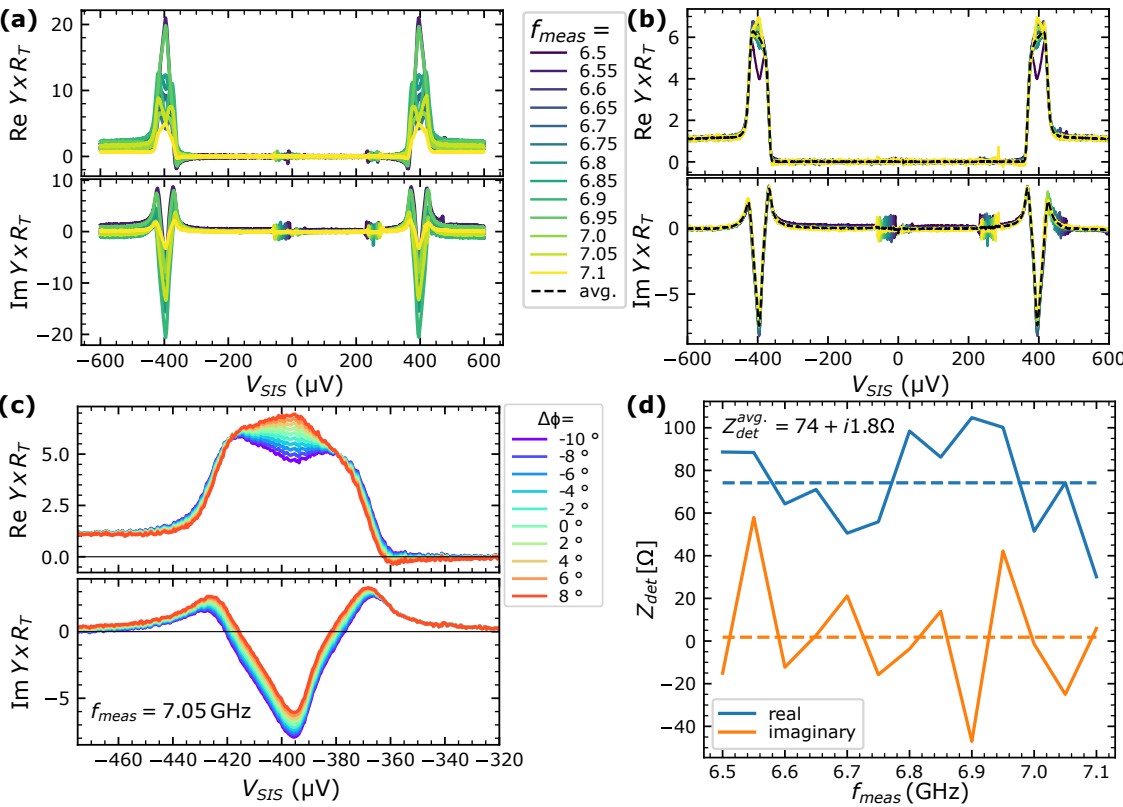

Figure 6: **(a)** Admittances obtained by calibrating the gain of the chain $G(f)$ and assuming a constant $50\,\Omega$ detection impedance. **(b)** Admittances obtained by further assuming that the detection impedance is frequency dependent. The latter is calibrated assuming the admittance is purely real at $V_{SIS} = -225\,\mu V$ with the value given by BCS prediction: $\mathrm{Re}\,Y(eV_{SIS} = 3\Delta, hf/\Delta = 0.14) \simeq 1.1/R_T$. **(c)** Impact of the phase calibration on the admittance measured at $f_{meas} = 7.05\,$GHz. Few degrees strongly modify the shape of the coherence peak, yet average to the shape expected from BCS predictions. **(d)** Real (blue) and imaginary (orange) parts of the detection impedance obtained by imposing that the admittance measured at $V_{SIS} = -600\,\mu V$ matches BCS predictions. An oscillatory pattern with period $\Delta f \simeq 400\,$MHz (25 cm standing wave) seems to develop on the real part around an average of $74\,\Omega$.

wave which is about the physical distance between the sample and the bias tee. The average detection impedance plotted as dashed lines is essentially real $Z_{det}^{avg.} = 74 + i1.8\,\Omega$. In Figure 6 (b), we show the admittances obtained by using this calibrated detection impedance in Eq. (9). The resulting admittance curves measured at different frequencies collapse at the largest bias, by construction, and their overall shape is now much more similar, which seems to validate our approach.

However, there remains a systematic difference among the admittances measured within the coherence peak $eV_{SIS} \simeq 2\Delta$. To understand this effect, we focus on biases close to $eV_{SIS} = 2\Delta$ for the curve measured at $f_{meas.} = 7.05\,\mathrm{GHz}$ which is shown in Figure 6 (c). The curve obtained from our calibration is compared to the ones obtained from the same calibration but adding a small phase factor $e^{2i\pi\Delta\phi}$ to the gain. We observe the shape $\mathrm{Re}\,Y$ around $eV_{SIS} = 2\Delta$ is extremely sensitive to phase estimation errors, which could be expected since the imaginary part peaks quite sharply at this bias. Nevertheless, the deviations at small positive and negative phases compensate each other. The data shown as a dashed line in Figure 6 (b) is the average of the admittances measured at the 13 different frequencies shown in the same plot. The averaging ensemble seems to be large enough to compensate our small phase calibration errors. The data $Y_{VNA}$ reported in Figure 4 of the main text is this averaged admittance.

## C.2 BCS prediction

The experimental protocol is based on BCS predictions of the admittance at $eV_{SIS} = 3\Delta$. In order to compute it, we exploit the non-equilibrium fluctuation-dissipation relations of tunnel junctions [35, 38, 43] linking the real part of the admittance to the DC $I(V)$ curve:

$$\mathrm{Re}\,Y_{BCS}(V_{SIS}, f) = \frac{I_{BCS}(V_{SIS} + hf/e) - I_{BCS}(V_{SIS} - hf/e)}{2hf}.$$

Here $I_{SIS}(V_{SIS})$ is the DC current voltage of a SIS tunnel junction predicted from BCS theory. In the zero temperature and zero depairing limit it can be computed from special functions [37, 54]:

$$I_{SIS}(V_{SIS}) = \frac{\Delta}{eR_T}\left(2xE(x) - \frac{1}{x}K(m)\right), \quad \text{for } x > 1,$$

where $m = 1 - 1/x^2$ with $x = eV/2\Delta$ and where $K(m)$ and $E(m)$ are the complete elliptic integrals of the first and second kind.

The imaginary part of $Y_{BCS}$ is computed by numerically integrating the Kramers-Kronig relation linking it to the real part:

$$\mathrm{Im}\,Y_{BCS}(V, f) = \frac{1}{\pi}\mathcal{P}\int df' \frac{\mathrm{Re}\,Y_{BCS}(V, f)}{f - f'}.$$

The curves shown as dashed lines in Figure 4 and the value $\mathrm{Re}\,Y_{CBS} \simeq 1.1/R_T$ at $V_{SIS} = -600\,\mu\mathrm{V}$ and $f = 6.8\,\mathrm{GHz}$ are obtained by evaluating these expressions at $hf/\Delta = 0.14$ and taking $\Delta = 200\,\mu\mathrm{eV}$.

# D  Power measurements and calibrations

## D.1  Raw excess power data

For completeness, we show in Figure 7 the raw data at the origin of Figure 2 in the main text. This raw data $\Delta P/k_B T_{amp}\Delta f$ is the excess power with respect to the noise of the amplification chain $\Delta P = P(V_{SIS}, V_{NIN}) - k_B T_{amp}\Delta f$ with $k_B T_{amp}\Delta f = P(V_{SIS} = V_{NIN} = 0)$. We always

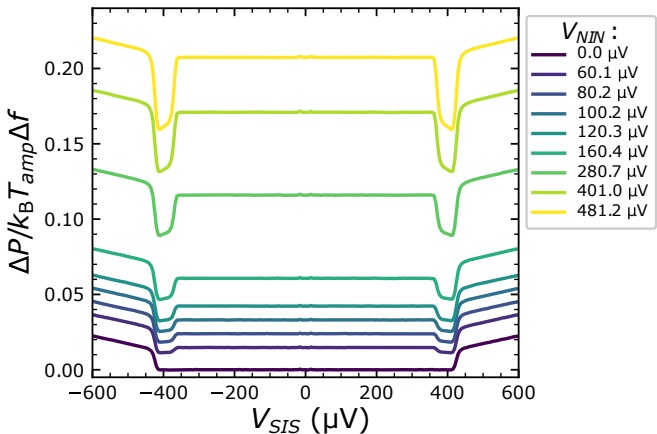

Figure 7: Excess power $\Delta P = P(V_{SIS}, V_{NIN}) - k_{\mathrm{B}}T_{amp}\Delta f$ normalized by the noise–power of the amplification chain $k_{\mathrm{B}}T_{amp}\Delta f = P(V_{SIS} = V_{NIN} = 0)$. The data shown in the main text Figure 2 is obtained by these curves subtracting the reference value at $V_{SIS*} = -225\,\mu\mathrm{V}$: $\Delta P_{ech}(V_{SIS}, V_{NIN}) = \Delta P(V_{SIS}, V_{NIN}) - \Delta P(V_{SIS}^*, V_{NIN})$.

normalize power measurements to the noise of the chain to correct for the small thermal drifts of the gain of the room–temperature RF amplification stage. The gain drifts are such that the measured $k_{\mathrm{B}}T_{amp}\Delta f$ drifts by about 1% over an hour; however power measurements are recorded every few seconds. The working hypothesis made, that whereas the room temperature gain drifts the noise $T_{amp}$ at its origin remains stable, is justified by the fact our recorded data is much more stable than the $k_{\mathrm{B}}T_{amp}\Delta f/100$ level as can be clearly seen in Figure 7 or 8.

The raw data shown in Figure 7 shows that even though the NIN junction strongly populates the line at large $V_{NIN}$ biases, the excess power remains constant for $|eV_{SIS}| < 2\Delta - hf_0$ until the radiation can interact with BCS excitations. The only deviation from this trend is a small contribution, barely visible on the scale shown, of the AC Josephson effect at the coupled bandwidth $2eV_{SIS} \simeq hf_0 = 28\,\mu\mathrm{eV}$. Such small Josephson radiation peaks develop since we did not manage to fully cancel the Josephson coupling despite the application of the small magnetic field within the squid loop. Nevertheless, the raw data shows that for whatever value of $V_{SIS}$ such that $hf_0 < |eV_{SIS}| < 2\Delta - hf_0$ the SIS junction does not interact with the radiation in the line. We arbitrarily chose the value $V_{SIS}^* = -225\,\mu\mathrm{V}$ within this window as a reference for this noninteracting situation.

The exchanged power shown if Figure 2 of the main text is obtained from the excess power shown in Figure 7 by subtracting this reference point:

$$\frac{\Delta P_{exch}}{k_{\mathrm{B}}T_{amp}} = \frac{\Delta P(V_{SIS}, V_{NIN})}{k_{\mathrm{B}}T_{amp}} - \frac{\Delta P(V_{SIS}^*, V_{NIN})}{k_{\mathrm{B}}T_{amp}}.$$

## D.2 Occupation calibration

In this section, we describe the protocol used to calibrate the photon population reaching the SIS junction. It relies on calibrating the noise power measured by the detection chain when it is emitted by the shot-noise of the tunnel junctions in the normal state.

### D.2.1 Power attenuation coefficient of the detection chain

The first step in our calibration procedure is to calibrate the power attenuation coefficient, relating the power at the output of the tunnel junctions to the power reaching the input of the

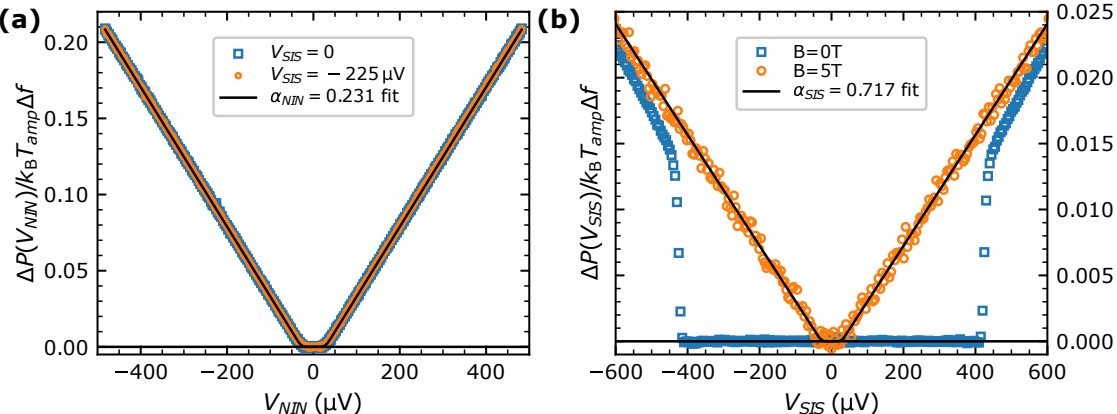

Figure 8: **(a)**: shot–noise excess power $P(V_{NIN}) - k_B T_{amp}\Delta f$ of the NIN junction when it is not interacting with the SIS junction. The data measured at $V_{SIS} = 0$ (blue squares) and $V_{SIS} = -225\,\mu V$ (orange circles) is indistinguishable: the blue circles were made larger (3pt) than the orange (2pt) for the curve to be visible. Both curves are well fitted with standard shot–noise theory of normal tunnel junction Eq. (D.1) calibrating for the attenuation coefficient $\alpha_{NIN} = 0.231$. **(b)**: shot–noise excess power of the SIS junction $P(V_{SIS}) - k_B T_{amp}\Delta f$ when there is no bias applied to the NIN junction (zero incomming mode population in the line). Blue squares is the excess power measured in the superconducting regime (already shown in Figure 7 for $V_{NIN} = 0$). Orange circles is the excess power in the normal state reached upon applying 5.56T to the junction. The curve measured in the normal state is well fitted with the standard shot–noise theory of tunnel junctions Eq. (D.1) calibrating for the attenuation coefficient $\alpha_{NIN} = 0.717$.

cryogenic amplifier.

Since normal tunnel junctions have bias-independent admittances in the RF domain (namely $Y(V, f) = Y(0, f)$), there is no difference in their emission and absorption *excess* noises

$$\Delta S_{II}(V, f) = \Delta S_{II}(V, -f),$$

defined as $\Delta S_{II}(V, \pm f) = S_{II}(V, \pm f) - S_{II}(0, \pm f)$. Therefore, both are equal to the excess symmetrized noise $(\Delta S_{II}(V, f) + \Delta S_{II}(V, -f))/2$ as well. Because of this, the excess noise of normal tunnel junctions can be safely used to calibrate the gain of the detection chain between the junction and the amplifier [25–27, 40] without the need to worry about a proper theory of power detection.

We further assume that the normal tunnel junctions behave as linear conductors. This is a good approximation for circuits with an impedance lower than the resistance quantum [21, 55], which is the case of our experiment since the impedance to ground is shunted by the $50\,\Omega$ detection circuit. With this approximation the excess power detected is that of a current source of noise power $\delta I^2 = (\Delta S_{II}(V, f) + \Delta S_{II}(V, -f))\Delta f = 2\Delta S_{II}(V, f)\Delta f$ in parallel with both the output impedance of the tunnel junction $R_T$ and the input detection impedance $Z_{det}$.

This approximation gives the classical result for the excess power coupled to the detection circuit:

$$\frac{\Delta P(V, f)}{k_B T_{amp}\Delta f} = \alpha \frac{(1 - |\Gamma|^2)R_T \delta I^2(V, f)}{4k_B T_{amp}\Delta f}$$

$$= \alpha \frac{(1 - |\Gamma|^2)R_T \Delta S_{II}(V, f)}{2k_B T_{amp}}.$$

(D.1)

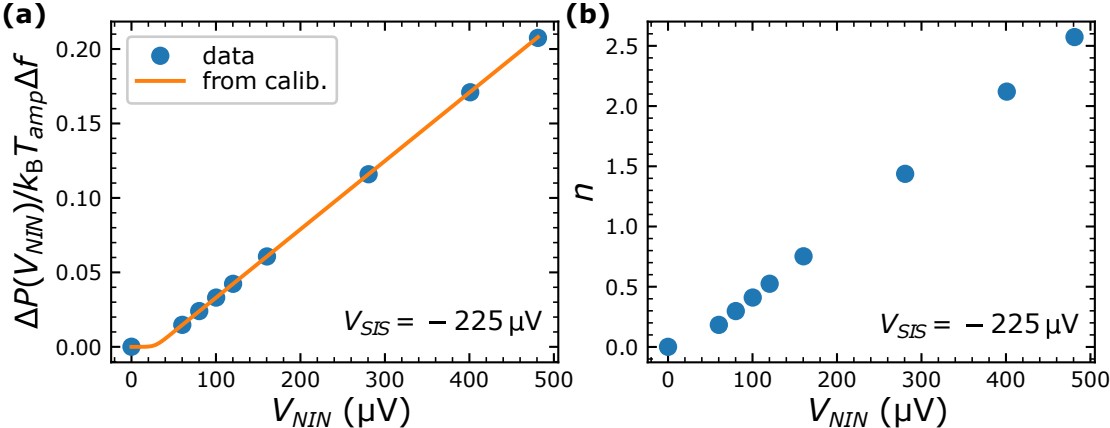

Figure 9: **(a)** Excess noise power taken from Figure 7 at $V_{SIS} = -225\,\mu V$ (blue dots), compared to the curve fitting the data shown in Figure 8 (a) (orange curve). Both are measurements of the same quantity, but measured 3 days apart. Their agreement demonstrates the stability of the added noise by the cryogenic amplifier. **(b)** Calibrated photon occupation at the input of the SIS junction when it is not interacting with the superconducting quasiparticles $eV_{SIS} = -225\,\mu V$. It is obtained by assuming that the corresponding microwave power is perfectly reflected by the SIS junction, and attenuated by the power attenuation factor $\alpha_{SIS}$ calibrated in Figure 8 (b).

In this expression, $\alpha$ accounts for the power attenuation between the source and the amplifier. The term $(1 - |\Gamma|^2)$ is the RF power transmission coefficient between the noise source and its detection circuit defined with the reflection coefficient $\Gamma = (R_T - Z_{det})/(R_T + Z_{det})$. Finally, $\Delta S_{II}(V, f)$ is the finite frequency excess noise power of the tunnel junction (with a negligible energy dependence on the electronic scattering amplitudes) [3]:

$$R_T \Delta S_{II}(V, f) = \sum_{\pm} \frac{hf \pm eV}{e^{(hf \pm eV)/k_B T} - 1} - \frac{2hf}{e^{hf/k_B T} - 1}.$$

Note that the detection bandwidth cancels in Eq. (D.1) since we neglect any frequency dependence of the spectral densities in the coupled bandwidth $\Delta f / f_0 \simeq 0.1$. We stress the consistency of Eq. (D.1) with Lesovik and Loosen formula Eq. (2) in the main text. Indeed, when evaluated in the classical source regime $S_{II}(f) = S_{II}(-f)$, Lesovik and Loosen formula reduces to the classical expression Eq. (D.1) taken in the strong impedance mismatch limit $1 - |\Gamma|^2 \simeq 4 \frac{\text{Re } Z_{det}}{R_T}$ where Lesovik and Loosen formula applies.

Figure 8 (a) shows the excess power of the NIN junction measured for $V_{SIS} = 0$ (blue squares) and $V_{SIS} = -225\,\mu V$ (orange circles), namely in biases for which the radiation is not coupled to the quasiparticles of the SIS junction (see Figure 7 and the corresponding discussion in the previous subsection). Both curves are indistinguishable and can be fitted with Eq. (D.1) giving an RF attenuation coefficient $\alpha_{NIN} = 0.231$ assuming a detection impedance $Z_{det} = 50\,\Omega$, a tunneling resistance $R_T^{NIN} = 41.8\,\Omega$ as measured independently, an added noise temperature by the chain of $T_{amp} = 2.9\,K$ and an electronic temperature of $T_{elec.} = 30\,mK$ accounting for the thermal rounding around $eV = hf_0 = 28\,\mu eV$.

Figure 8 (b) shows the excess power of the SIS junction measured in the superconducting state (as already shown in Figure 7 for $V_{SIS} = 0$) in blue squares. It is compared with the excess power measured in the normal state (orange circles) reached upon applying a magnetic field $B = 5.56\,T$ larger than the critical field of Aluminum $\simeq 200\,mT$. The curve measured in the normal state can be fitted with Eq. (D.1) giving an RF attenuation coefficient $\alpha_{NIN} = 0.717$

assuming a detection impedance $Z_{det} = 50\,\Omega$, a tunneling resistance measured independently in Appendix E $R_T^{SIS} = 6712\,\Omega$, an added noise temperature by the chain of $T_{amp} = 2.9\,\text{K}$ and an electronic temperature of $T_{elec.} = 30\,\text{mK}$ accounting for the thermal rounding around $eV = hf$. It can be seen that the noise power emitted in the superconducting state tends to the one emitted in the normal state for biases larger than $2\Delta$, yet the slopes at the largest measured bias are still different. This is typical of superconducting tunnel junctions where transport features associated to the coherence peak of the BCS density of states peak decay (algebraically) slowly for energies larger than the gap. For the same reason, the value taken by the real part of the admittance at $eV_{SIS} = 3\Delta$ used in the admittance calibration is similar to the DC tunneling resistance, but not exactly the same: $\text{Re}\,Y(eV_{SIS} = 3\Delta, hf/\Delta = 0.14) \simeq 1.1/R_T$.

### D.2.2  Calibration of the photon population

With these calibrations in hand, we can now predict the occupation number $n(V_{NIN})$ of the radiation seen by the SIS junction in the coupled bandwidth. The average power driven through a transmission line reads $\langle \hat{P} \rangle = \int df\, hf\,(n^{\rightarrow}(f) - n^{\leftarrow}(f))$, with $n^{\leftrightarrows}(f)$ the average occupation number of the right and left movers. At the input of the amplifier the left movers are populated by the constant amplifier noise ($n^{\leftarrow}(f) = k_B T_{amp}/hf$ in the coupled band). The right movers power is fed from the output of the SIS junction but suffers from the power attenuation factor $\alpha_{SIS}$ calibrated in the previous section. By tuning the SIS junction bias to $V_{SIS}^* = -225\,\mu\text{V}$, such that the radiation does not interact with the superconducting quasiparticles (see discussion on the raw excess power data), the SIS junction simply reflects the occupation number it sees at its input $n(V_{NIN})$. Therefore, the excess power $\Delta P(V_{NIN}, V_{SIS}^*) = P(V_{NIN}, V_{SIS}^*) - P(V_{NIN} = 0, V_{SIS}^*)$ reads:

$$\Delta P(V_{NIN}, V_{SIS}^*) = hf_0 \alpha_{SIS} n(V_{NIN}) \Delta f \,,$$

where we neglected the small frequency dependence of the power emitted by the *NIN* junction in the coupled bandwidth $\Delta f$. Rearranging terms and introducing a trivial factor $k_B T_{amp}/k_B T_{amp}$ we obtain a relation between the occupation number $n(V_{NIN})$ and previously measured or calibrated quantities:

$$n(V_{SIS}) = \frac{\Delta P(V_{NIN}, V_{SIS}^*)}{k_B T_{amp} \Delta f} \frac{k_B T_{amp}}{\alpha_{SIS} hf_0} \,. \tag{D.2}$$

The occupation number resulting from this calibration is shown in Figure 9 (b) as a function of the NIN voltage bias, for the values of NIN voltage used to produce figures 2 and 3. It is a prediction based on the measurement of $\frac{\Delta P(V_{NIN}, V_{SIS}^*)}{k_B T_{amp} \Delta f}$ shown in Figure 9 (a), and the calibration of $\alpha_{SIS}$ performed in the previous section with the data shown in Figure 8 (a).

We would like to stress that the first term on the right–hand side of Eq. (D.2) is the excess power data shown in Figure 7 taken at the value $V_{SIS}^* = -225\,\mu\text{V}$. One can further see that the noise added by the line and the power attenuation coefficient appear through the ratio $\frac{k_B T_{amp}}{\alpha_{SIS}}$. The precise value used for $k_B T_{amp}$ will not change the calibration of the occupation procedure as soon as it is the same as used to calibrate the power attenuation coefficient through the Eq. (D.1). The important point is that the noise referred to the input of the amplifier of the line remains stable. Figure 9 (a) demonstrates the stability of the chain by comparing measurements performed 3 days apart: The blue dots are the excess power measured at $V_{SIS}*$ for the set of data shown in Figure 7, and used for the calibration of the occupation number shown in Figure 9 (b) exploiting Eq.(D.2). The excess noise is compared with the orange curve that was obtained by fitting the data shown in Figure 8 (a) measured 3 days later. Both curves agree within a one–percent range factor, demonstrating the stability of the added noise temperature of the amplifier.

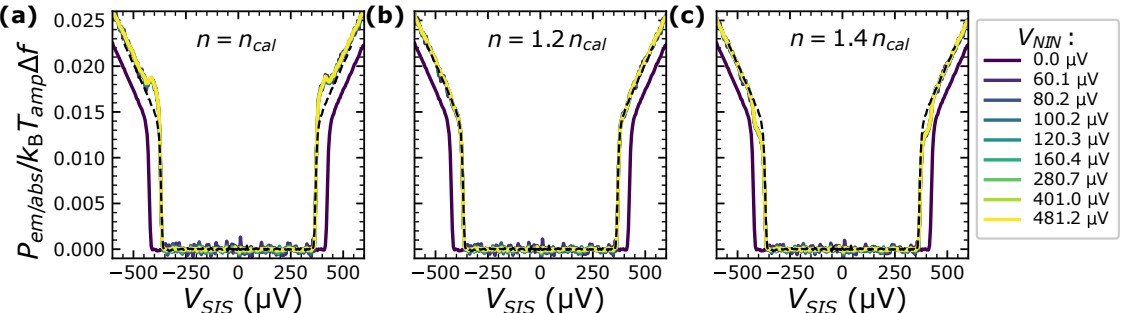

Figure 10: Comparison of emission ($V_{NIN} = 0$) and absorption ($V_{NIN} \neq 0$) noise curves obtained through the methodology explained in the main text, but changing the population of the RF modes by an overall scale factor. **(a)** the population is as dictated by the photon population section $n(V_{NIN}) = n_{cal}(V_{NIN})$ and shown in Figure 9 (b). **(b)** the population is scaled by a 1.2 factor $n(V_{NIN}) = 1.2\, n_{cal}(V_{NIN})$. **(c)** the population is scaled by a 1.4 factor $n(V_{NIN}) = 1.4\, n_{cal}(V_{NIN})$. The black dashed line is the absorption noise curve predicted from the emission noise curve via Rogovin and Scalapino predictions [35]. This prediction is thus insensitive to overall scale factors in the population.

### D.2.3 Ad-hoc population correction

As mentioned in the main text, despite our calibration efforts we had to correct the population by a factor of 1.2 (0.8 dB). Our occupation number calibration procedure neglects spurious wave reflection in the path between the NIN and the SIS junctions. Yet as shown in the admittance calibration section Appendix C, our coherent reflectometry measurements showed non–negligible standing wave effects arising from such reflections. Unfortunately, we cannot exploit these measurements because the corresponding propagation paths are not the same.

Figure 10 (a) shows the emission and absorption noise power extracted from the exchanged power measurements using the methodology explained in the main text, but now using the photon occupation as dictated by the calibration shown in Figure 9 (b). The absorption noise curves are compared with the black dashed line predicted from the measured emission noise through Rogovin and Scalapino relations [35]. In the large bias limit $eV \gg k_B T$ they take the simple form:

$$P_{abs}(V, f) = P_{em}(V + \text{sign}(V)2hf/e, f),$$

where $\text{sign}(V)$ is the function returning the sign of variable $V$. Although initially derived for superconducting tunnel junctions from microscopic considerations [35], such relations generically hold [38, 43, 56] for tunneling conductors if co-tunneling effects [42] and tunneling dwell–time [57] are negligible. This prediction is *immune to any calibration error*: it only assumes that the exchanged noise–power measured when there is no incoming field in the line is given by the emission noise. The data shown in Figure 10 (a) gives an overall fair agreement between the absorption noise we would predict from Rogovin and Scalapino formula and the absorption noise extracted from our protocol. Nevertheless, we observe a 10% overshoot around the coherence peak, which hints at a calibration error. Figure 10 (b) shows that an overall scale correction of 20% in the calibration gives an excellent agreement, whereas Figure 10 (c) shows that a scale correction of 40% decreases the agreement. The data shown in Figure 3 of the main text corresponds to this 20% scale correction to the calibrated population.

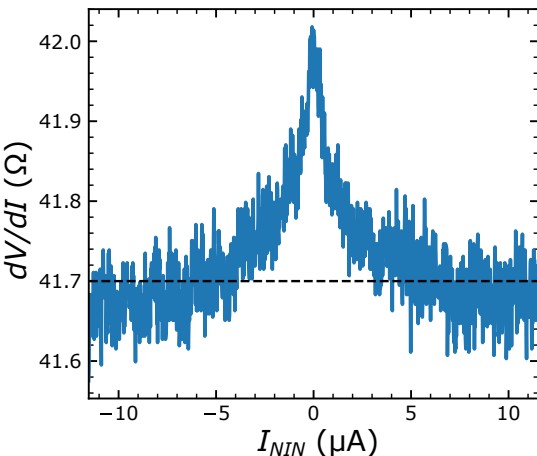

Figure 11: Differential resistance $dV/dI$ of the NIN tunnel junction measured as a function of the DC current $I_{NIN}$ feeding it. It displays a variation smaller than 1% in the measured range (about $\pm 600\,\mu V$). The black dashed line is the average value we use to convert DC current into DC bias and to predict the shot–noise it emits.

We would like to stress that an error in population calibration changes the shape of the absorption noise extracted from the exchanged power via Eqs. (4) and (5) in the main text as manifest in Figure 10. However, the difference $P_{abs} - P_{em}$ used to build the real part of the admittance via Kubo formula only depends on $n(V_{NIN})$ via a global scale factor. This is the very meaning of Kubo relation: i) when the admittance is positive it describes Joule heating, namely the magnitude of the power dissipated in the conductor is proportional to the external power $n$ and to the difference of absorption and emission noise. ii) If the admittance is negative then the difference $P_{abs} - P_{em}$ is negative as well, and Kubo relation describes the gain of the system in the linear regime [46].

# E DC characterization

The patient and careful reader having reached this part of the appendices will have realized that our calibration methods are based on the knowledge of the DC resistance of the tunnel junctions which determines both the RF coupling to the detection circuit but also the magnitude of the RF shot–noise used to calibrate the lines.

## E.1 NIN junction

The NIN junction is current–biased via a 1 MΩ polarization resistor set at room temperature with a low frequency single tone of frequency 10 Hz and amplitude such that the rms voltage fluctuations at the input of the junctions are about $1\,\mu V_{rms}$ on top of the DC current bias. The polarization line is low–pass filtered with second–order RC filters thermally anchored at the MC chamber stage. The voltage drop at the input of the junctions is measured via synchronous lock-in detection. The resulting $dV/dI(I)$ is shown as a blue curve in Figure 11 giving an average resistance $R_{NIN} = 41.7\,\Omega$ shown as a dashed line.

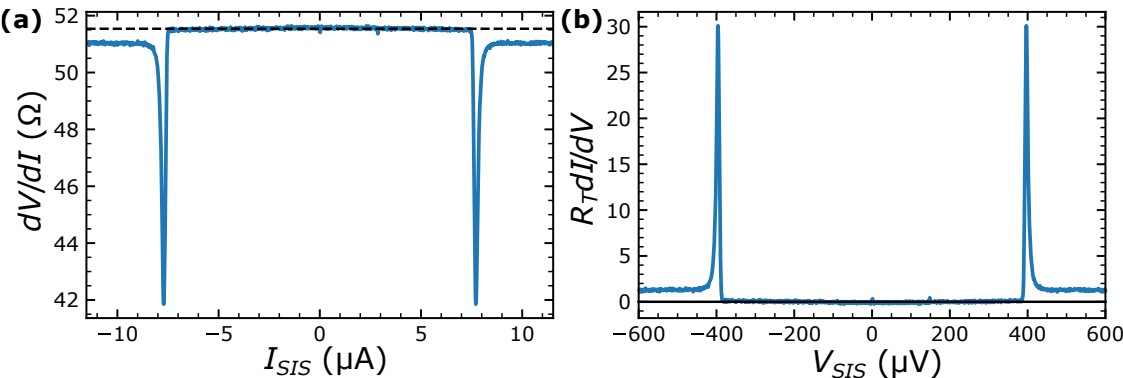

Figure 12: **(a)** Differential resistance $dV/dI$ of the SIS tunnel junction in parallel of its shunting resistor measured as a function of the DC current $I_{SIS}$ feeding them. **(b)** Differential conductance $dI/dV$ of the SIS junction as a function of the DC voltage applied $V_{SIS}$. It is obtained by taking the inverse of the differential resistance and subtracting the background conductance $1/51.5\,\Omega$ shown as a dashed line in (a). The SIS voltage is obtained by numerical integration of the curve in (a).

## E.2 SIS junction

### E.2.1 Voltage polarization setup

The calibration of the SIS tunnel junction is more engaged since we use a 50 Ω thin film resistor to convert the polarization current into a polarization voltage $V_{SIS}$ at the junction input, as shown in Figure 1 of the main text.

Figure 12 (a) shows the differential resistance $dV/dI$ in blue of the parallel composition of the SIS and its shunting resistor. It shows a marked superconducting behavior on top of a background resistance $R_{shunt} = 51.5\,\Omega$ (black dashed line). The curve is numerically integrated to obtain the applied DC bias $V_{SIS}$. The SIS differential conductance is obtained by inverting the differential resistance and subtracting a constant background conductance $1/R_{shunt}$. It is shown in Figure 12 (b), where superconducting coherence peaks are manifest at $eV_{SIS} = \pm 400\,\mu$V.

The background conductance within the superconducting transport gap is nevertheless slightly non-linear even though it is not related to the physics of the SIS junction: i) we observe a similar behavior at strong magnetic fields applied to the SIS where superconducting features are absent, and ii) we do not observe any dissipative behavior in the energy exchanges within the gap besides the Josephson peaks. Because of this non-linear background, the precise value of the tunneling resistance obtained by this DC characterization is strongly impacted by experimental biases: A difference in the **fourth digit** of the shunting resistance gives a 10% difference of the resulting tunneling resistance. This is due to the extremely unfavorable current division ratio of the voltage polarization scheme. On the other side, the voltage obtained by numerical integration of the $dV/dI(I)$ curve agrees within the 1% range with the voltage given by $R_{shunt}I_{SIS}$. For these reasons, we **do not** make any quantitative claim from these measurements other than the value of the superconducting gap $\Delta$ and the values of the onset of emission and absorption noises at $|eV_{SIS}| = 2\Delta \pm hf_0$.

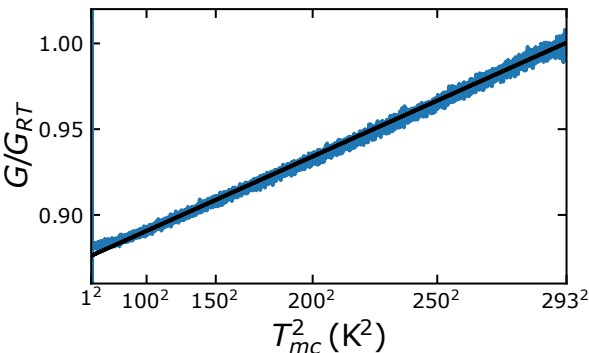

Figure 13: Blue curve: Temperature evolution of the SIS junction conductance $G = (dV/dI)^{-1}$ normalized by its room temperature value. This measurement was performed during a dedicated run where the shunting resistor was removed. The mixing chamber temperature is recorded on the Cernox thermometer provided by the vendor of the dilution fridge. The black curve is the expected [58] $G(T) = G_0(1 + (T/T_0)^2)$ evolution.

### E.2.2   Current polarization setup

In order to have an independent estimate of the tunneling resistance of the SIS junction, we performed a dedicated run where the shunting resistance was removed. This way, the SIS junction is directly current biased and the corresponding voltage drop is measured in a 3 point measurement similar to the one performed on the NIN junction.

Figure 13 shows as a blue curve the conductance of the SIS junction $G = (dV/dI)^{-1}$ as a function of the mixing chamber temperature measured by the Cernox thermometer provided by the dilution fridge vendor. The tunneling conductance is normalized to its room temperature value. We measured a $\simeq$ 13% decrease in the tunneling conductance between the room temperature measurement and the one performed at 1 K. Moreover, the temperature dependence of the tunneling conductance is very well described by the expected [58] temperature evolution $G(T) = G_0(1 + (T/T_0)^2)$ used to build the black curve. Because of this agreement (giving an uncertainty within 1% for the low temperature value) and the way better accuracy in terms of the DC measurement, we use this evolution with the temperature as a calibration of the tunneling conductance.

Just before the cooldown leading to all data shown in this article (besides Figure 13), we measured at room temperature a DC resistance of 5940 Ω. Applying the 13% increase expected from the temperature dependence shown in 13 we obtain the value $R_T = 6712\,\Omega$ which is used throughout the article.

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
