# Peer review of "Testing Kubo formula on a nonlinear quantum conductor driven far from equilibrium via power exchanges"

_SciPost Physics, doi:SciPost Phys. 19, 127 (2025)_

## Round 1 · Referee Report · Anonymous (Referee 1) · 2025-7-11

Strengths

  • The emission from a superconducting circuit out of equilibrium is a timely topic.
  • Investigating the fluctuation-dissipation relation in far-from-equilibrium regimes is relevant to a wide audience.
  • The manuscript offers a nice interplay between theoretical and experimental results.

Report

In the manuscript, the authors test Kubo’s formula in a far-from-equilibrium regime. To separately extract the emission and absorption noise, they implement the idea of Lesovik and Loosen by coupling the device under test to a linear circuit with a large impedance mismatch. By varying the effective temperature of the latter, they are able to distinguish the zero-temperature emission spectrum from the absorption spectrum. The impedance of the circuit—representing its linear response around the strongly non-equilibrium state—has been independently determined using standard microwave techniques. The central finding of the paper is presented in Figure 4, which provides an experimental test of Kubo’s formula.

Overall, the paper is clearly written, and the results are both timely and relevant. The manuscript easily meets the standards of the journal and should be accepted with only minor revisions.

Requested changes

I have only a few small things that I would like the authors to clarify:

  • line 35: what do you mean with " the noise arising from quantum fluctuations is non-causal"? Causality is a concept that I do not really understand in the context of fluctuations...
  • line 80: it is not clear what is ment by "This lower order expression is well justified". Do you mean "This approximate expression" or "This perturbative expression"?
  • line 88: for clarity "coupled circuit" would be better called "detection circuit"
  • in the abstract " in the good current source limit " is not completely clear; maybe "acts as a current source for the detection circuit"
  • line 332: "Developing it to lowest" could be clearer formulated as "Expanding it to lowest"

Recommendation

Publish (surpasses expectations and criteria for this Journal; among top 10%)

  • validity: top
  • significance: top
  • originality: top
  • clarity: high
  • formatting: perfect
  • grammar: good

Author:  Carles Altimiras  on 2025-10-08  [id 5902]

(in reply to Report 1 on 2025-07-11)

We thank the referee for their positive report; their remarks contributed to improve the quality of the manuscript. We have implemented the requested changes in the resubmitted version.

---

## Round 1 · Referee Report · Anonymous (Referee 2) · 2025-7-29

Strengths

1) Clear goal of the research, experimental verification of a sound theoretical concept

Weaknesses

1) The manuscript needs proofreading and possibly some clarifications.

Report

The article presents an experimental test of the Kubo formula in an SIS junction which functions as a quantum conductor. A perfect comparison between theory and experiment is achieved, and the authors provide solid theoretical basis for the results. They also consider the situations far out of equilibrium, and take into account the backaction on the conductor. This is a very useful piece of work which certainly deserves publication.

Requested changes

I only have minor comments to the manuscript:

1) Where does Eq. (D.2) come from? It seems essential for calibration of the number of photons. Either a reference, or a hint at a derivation are needed.

2) A related question: A number of photons in Fig. D.3 is small and never exceeds 3. There is no problem with it per se as soon as the authors have enough points, but I guess the precision would be higher if a greater range of n would be used. Did the authors explore this (at different voltages)?

3) The manuscript must be proofread, there are quite some typos, and Refs. 3 and 56 are the same.

Recommendation

Publish (surpasses expectations and criteria for this Journal; among top 10%)

  • validity: high
  • significance: high
  • originality: top
  • clarity: high
  • formatting: perfect
  • grammar: good

Author:  Carles Altimiras  on 2025-10-08  [id 5903]

(in reply to Report 2 on 2025-07-29)

We thank the referee for their positive report; their remarks contributed to improve the quality of the manuscript. We detail below how we addressed the specific comments and request:

1/ We agree with the referee the manuscript was missing a derivation of Eq. D.2. We have changed the first paragraph of section D.2.2 to provide it now.

2/ The referee is right that the absorption noise extracted from our protocol has a better precision when increasing the occupation number. This fact can be clearly seen in the data shown in Fig. 3, especially at low SIS biases. However, exploring larger occupations/NIN biases would not help much since i) the accuracy achieved for the absorption noise at the largest NIN bias is similar to the one we obtained for the emission noise. ii) the occupation numbers we used already provide a far better precision than our accuracy, which is limited by the spurious RF return losses neglected by our calibration protocol as stressed in section D.2.3. We added a sentence in line 542, clarifying that the occupation number shown in Figure D.3 is a prediction based on other measurements and calibrations.

3/ We corrected the problem with the references and proofread the manuscript thoroughly. We tried to correct the typos and the grammar as much as possible.

---

## Round 2 · Author Response

We thank the referees for their positive report; their remarks contributed to improve the quality of the manuscript. We detail below how we addressed the specific comments and request:

---

## Round 2 · List of Changes

Answers to report #2: 1/ We agree with the referee the manuscript was missing a derivation of Eq. D.2. We have changed the first paragraph of section D.2.2 to provide it now. 2/ The referee is right that the absorption noise extracted from our protocol has a better precision when increasing the occupation number. This fact can be clearly seen in the data shown in Fig. 3, especially at low SIS biases. However, exploring larger occupations/NIN biases would not help much since i) the accuracy achieved for the absorption noise at the largest NIN bias is similar to the one we obtained for the emission noise. ii) the occupation numbers we used already provide a far better precision than our accuracy, which is limited by the spurious RF return losses neglected by our calibration protocol as stressed in section D.2.3. We added a sentence in line 542, clarifying that the occupation number shown in Figure D.3 is a prediction based on other measurements and calibrations. 3/ We corrected the problem with the references and proofread the manuscript thoroughly. We tried to correct the typos and the grammar as much as possible.

Answers to report #1: We thank the referees for their positive report; their remarks contributed to improve the quality of the manuscript. We detail below how we addressed the specific comments and request: Line 35: what we had in mind is that classically one could in principle track all the microscopic degrees of freedom of the physical system and predict the time evolution of a physical signal, which might otherwise look like a fluctuating stochastic signal to an observer not having this information. However, one cannot find such a causal origin to the time dependence of quantum fluctuating signals. Nevertheless, we agree with the referee this is not a standard definition, and decided to remove the reference to causality in the new version. Line 80: We implemented the suggested change, it now reads : “This perturbative expression …” Line 88: We implemented the suggested change, it now reads : “… into the detection circuit …” Abstract: We implemented the suggested change, it now reads : “… the conductor acts as a current source” Line 332: We implemented the suggested change, it now reads : “Expanding it to lowest order in…”

---

## Editorial Decision

published